# Signaling and Resistosome Formation in Plant Innate Immunity to Viruses: Is There a Common Mechanism of Antiviral Resistance Conserved across Kingdoms?

**DOI:** 10.3390/ijms241713625

**Published:** 2023-09-03

**Authors:** Peter A. Ivanov, Tatiana V. Gasanova, Maria N. Repina, Andrey A. Zamyatnin

**Affiliations:** 1Faculty of Biology, Lomonosov Moscow State University, Moscow 119234, Russia; regaflight@gmail.com (P.A.I.); tv.gasanova@gmail.com (T.V.G.); rep-masha@yandex.ru (M.N.R.); 2Faculty of Bioengineering and Bioinformatics, Lomonosov Moscow State University, Moscow 119234, Russia; 3Belozersky Institute of Physico-Chemical Biology, Lomonosov Moscow State University, Moscow 119992, Russia; 4Research Center for Translational Medicine, Sirius University of Science and Technology, Sirius 354340, Krasnodar Region, Russia; 5Institute of Translational Medicine and Biotechnology, Sechenov First Moscow State Medical University, Moscow 119991, Russia

**Keywords:** plant virus, nucleotide-binding leucine-rich proteins, hypersensitive response, resistosome, programmed cell death

## Abstract

Virus-specific proteins, including coat proteins, movement proteins, replication proteins, and suppressors of RNA interference are capable of triggering the hypersensitive response (HR), which is a type of cell death in plants. The main cell death signaling pathway involves direct interaction of HR-inducing proteins with nucleotide-binding leucine-rich repeats (NLR) proteins encoded by plant resistance genes. Singleton NLR proteins act as both sensor and helper. In other cases, NLR proteins form an activation network leading to their oligomerization and formation of membrane-associated resistosomes, similar to metazoan inflammasomes and apoptosomes. In resistosomes, coiled-coil domains of NLR proteins form Ca^2+^ channels, while toll-like/interleukin-1 receptor-type (TIR) domains form oligomers that display NAD+ glycohydrolase (NADase) activity. This review is intended to highlight the current knowledge on plant innate antiviral defense signaling pathways in an attempt to define common features of antiviral resistance across the kingdoms of life.

## 1. Introduction

Programmed cell death (PCD) is a fundamental aspect of functioning of multicellular organisms. At the morphological and biochemical levels, there are a number of basic similarities, even among representatives of different kingdoms of living organisms (plants and animals), although the molecular mechanisms of programmed cell death vary. For example, classical apoptosis is a characteristic exclusive to animals, and caspases that are responsible for the development of this type of programmed cell death are not found in living organisms outside the animal kingdom [1]. The similarity between the animal and plant kingdoms is manifested at the level of alternative death processes, such as autophagy and ferroptosis [2,3,4,5]. The caspase-like activity observed in plants is attributable to metacaspases, vacuolar processing enzymes (VPE), and the papain-like cysteine proteases [6,7,8]. Protein substrates that are cleaved during the development of PCD in plants and animals may be orthologs. DNA fragmentation is also characteristic of many types of PCD in organisms belonging to different kingdoms. PCD in plants can be associated with their development (developmental PCD, dPCD) or with the invasion of a pathogen (pathogen-related, pPCD) [9]. The latter is often termed the hypersensitive response (HR). Despite the considerable amount of existing experimental data, the mechanism of PCD in plants remains insufficiently studied [9]. The induction of cell death in plants may depend on pathogen proteins called avirulence factors (Avrs). Virus-specific proteins occupy a special place among these. Viral effectors are variable, relatively small, and—in many cases—well-studied proteins that usually accumulate in plants to high levels. Experimental infections allow rapid testing of wild-type/mutant effectors and HR sensors [10]. During HR, cells infected by a pathogen undergo PCD, forming necrotic lesions on the plant.

## 2. Virus-Specific Proteins Involved in the Hypersensitive Response

Avr recognition by resistance proteins (R proteins) leads to a strong defense response. As a rule, virus-specific avirulence factors interact with the coiled-coil nucleotide-binding leucine-rich repeats (LRR) protein CC-NB-LRR (CNL) class and toll-like receptor/Interleukin (IL)-1 receptor nucleotide-binding LRR protein TIR-NB-LRR (TNL) class [6,9]. There are two paths for triggering an antiviral defense response: Avrs can bind to an R protein with or without the participation of an intermediate host factor; it is also possible for Avrs to induce the synthesis of an R gene product and allow it to signal a response. Interaction between viral and host factors leads to a rapid response [10]. Information concerning numerous viral proteins involved in triggering HR is summarized in Table 1.

### 2.1. Virus RdRp Proteins

#### 2.1.1. Tobacco Mosaic Virus (TMV) p50

The HR is most well studied for TMV infection. The *N*-gene product in *Nicotiana glutinosa* and replication-associated protein p50 (helicase domain of RNA-dependent RNA polymerase, RdRp) activate ATP-dependent HR induction [11,12,13,14,15,16,17].

TMV p50 interacts indirectly with toll-like/interleukin-1 receptor-type domain protein encoded by the *N* gene; in the *Nicotiana benthamiana,* such signaling requires an additional chloroplastic rhodanese sulfurtransferase, NRIP1 (Nuclear Receptor Interacting Protein 1), that recognizes p50 and inhibits the development of infection [18].

Squamosa Promoter Binding Protein Like 6 (SPL6) transcription factor also stimulates TMV resistance [19].

*N* Requirement Gene partner 1 (NRG1) participates in the plant response at the next step of signaling by forming a Ca^2+^ channel in the plasma membrane and promoting cation influx into the cell [20,21].

#### 2.1.2. Potyviral RdRp (NIb)

A known function of the Pepper Mottle Virus (PepMoV) nuclear inclusion protein b (NIb) is activation of HR with the participation of CNL-type proteins Pvr4 or Pvr9. The coiled-coil domain of Pvr4 might be involved in specific Avr recognition [22]. These proteins are capable of causing cell death in *Capsicum annuum* and *Capsicum chinense*. The Nib proteins of other potyviruses, pepper severe mosaic virus (PeSMV), tobacco etch virus (TEV), and potato Y virus (PVY), should also be mentioned [22,23,24,25].

#### 2.1.3. Plantago Asiatica Mosaic Virus (PlAMV) RdRp

The replicase of PlAMV is important for development of systemic necrosis. Infection of *N. benthamiana* with this potexvirus causes HR, while the PlAMV lily isolate Li6, in contrast to Li1, infects plants asymptomatically [26,27].

#### 2.1.4. Tomato Yellow Leaf Curl Virus (TYLCV) Rep/C1

Replication-associated protein Rep/C1 of begomovirus TYLCV contains the avirulence motif for binding to the *Ty-2* gene product in *Solanum habrochaites*. *Ty-2* encodes the NB-LRR protein. Interaction between Rep/C1 and Ty-2 triggers HR [28,29]. Expression of the N-terminal 210 amino acid residues of Rep/C1 in tomato activates the genes responsible for suppression of cell death [30,31].

#### 2.1.5. Cucumber Mosaic Virus (CMV) 2a

HR induced by CMV-replication-related protein 2a and TNL RS4-4 was observed in *Phaseolus vulgaris*. The efficiency of this interaction does not depend on the enzymatic activity of the replicase [32].

### 2.2. Virus Coat Proteins (CP)

#### 2.2.1. Potato Virus X (PVX) CP

The potato *Rx* gene encoding a CNL-class protein is responsible for HR during PVX infection [33]. The PVX CP is required for efficient interaction with the Rx protein, which causes necrosis in potato and local lesions in *Gomphrena globosa* [34]. Ran GTPase Activating protein 2 (RanGAP2) as well as transcription factor Golden-like (NbGLK1) from *N. benthamiana* enforce the relocalization of Rx from the cytoplasm into the nucleus [35,36,37].

#### 2.2.2. Turnip Crinkle Virus (TCV) CP

The coat protein of tombusvirus TCV interacts with the product of the *HRT* gene [38]. HRT-mediated resistance in the turnip is dependent on lipase-like protein Enhanced Disease Susceptibility 1 (EDS1) and its interacting lipase cofactor Phytoalexin Deficient 4 (PAD4), which form a complex with EDS1. A NB-LRR protein accomplishes downstream signaling [39,40]. The level of EDS1 and PAD4 synthesis is increased in the presence of sialic acid (SA), including exogenous treatment. In the latter case, HRT is stimulated via PAD4 [40,41,42,43].

#### 2.2.3. Potato Virus Y (PVY) CP

Potyvirus PVY infection of potato activates HR connected with CP PVY and *Ny* gene products [44,45,46,47].

#### 2.2.4. Cucumber Mosaic Virus (CMV) CP

The *RCY1* (CNL-class) gene product from *Arabidopsis thaliana* interacts with the CMV coat protein resulting in HR. *RCY1* and *HRT* belong to the same *HRT/RP8* family and apparently diverged via recombination. CPs of CMV and TCV are not homologous, but recognize allelic gene products [48,49].

### 2.3. Virus Movement Proteins (MP)

#### 2.3.1. Tomato Mosaic Virus (ToMV) MP

Movement proteins (MP) promote cell-to-cell viral transport through plasmodesmata. Resistance genes *Tm2*, *Tm2^2^* were found in tomato, *Solanum peruvianum*, *Nicotiana tabacum,* and *Chenopodium amaranticolor*. The *Tm2^2^* gene is MP-dependent, whereas the *Tm1* gene is activated by RdRp. *Tm2^2^* codes for a CNL protein, but the mechanism of HR induction is not fully understood [50,51,52,53].

#### 2.3.2. Tobacco Rattle Virus (TRV) MP

HR in tobacco and potato can be associated with the nucleus. TRV MP (29K) is the elicitor of HR response via the EDS1/PAD4 pathway. Overexpression of the 29K gene by agroinfiltration of leaves leads to cell death [46,54].

#### 2.3.3. Tomato Spotted Wilt Virus (TSWV) MP

*R* genes belonging to the *Sw-5* family from *Solanum peruvianum* code for CNL proteins. Tospovirus TSWV MP (NS_M_) induces cell death in tomato and *N. benthamiana* [55,56,57]. The RTSW gene product from *Nicotiana alata* also recognizes NS_M_ in tobacco. RTSW domains required for HR triggering are different from *Sw-5b*; most likely, the proteins are not orthologs [58].

#### 2.3.4. Cauliflower Mosaic Virus (CaMV) MP

The P6 protein of pararetrovirus CaMV induces systemic PCD in *Nicotiana clevelandii* and HR in *Nicotiana edwardsonii* [59]. P6 is also responsible for the non-necrotic defense response in *A. thaliana*, *Nicotiana bigelovii*, and *N. glutinosa* [59,60,61]. Transgene and transient expression of P6 in *Arabidopsis* and *N. benthamiana* changes the subcellular localization of transcription factor NPR1 (nonexpressor of pathogenesis-related genes 1) [62]. Introgression of *CCD1* gene from *N. bigelovii* can selectively block systemic and local HR induced by CaMV infection in *N. clevelandii* and *N. edwardsonii* [63].

### 2.4. Other Viral Proteins Involved in the HR

#### 2.4.1. Tomato Spotted Wilt Virus (TSWV) Silencing Suppressor Protein (NSs)

The NSs protein activates resistance of pepper and *C. chinense* against TSWV related to CNL *Tsw* resistance gene from tomato codes for a protein with unusually large LRR region (over 30 repeats) [64,65]. The NSs is a viral suppressor of RNA silencing [66] and a determinant for transmission by thrips (Thysanoptera) [67]. Changes at the N-terminus of NSs are crucial for interaction with Tsw sensor [68,69].

#### 2.4.2. Tomato Leaf Curl New Delhi Virus (ToLCNDV) Silencing Suppressor Protein AC4

*R*-gene *SISw5a* product from *Solanum lycopersicum* is a homolog of the Sw-5a CNL protein. SlSw5a protein binds to AC4 from ToLCNDV, which inhibits RNA interference. A mutation at the C-terminus of the AC4 protein destroys this interaction, thus preventing PCD [70,71].

#### 2.4.3. Soybean Mosaic Virus (SMV) Proteins CI, P3 and HC-Pro

Potyvirus SMV encodes CI (cylindrical inclusion), P3 and HC-Pro (helper component proteinase) proteins that can function as Avrs interacting with NB-LRR proteins. Genetic mapping studies identified two resistance loci (*Rsv* and *Rsc*) that code for typical CNL proteins which provide resistance to SMV [72,73]. P3/HC-Pro act as Rsv1, Rsv2, and Rsv4 inductors, while CI acts as a trigger for Rsc4-3 [72,73,74,75]. The interaction between an avirulent SMV strain and a soybean resistance gene product belonging to the Rsv family induced siRNA accumulation followed by autophagy [74]. P3 cooperates with translation factors to trigger the unfolded protein response (UPR) [75,76].

## 3. NLR Signaling Leading to Hypersensitive Response in Plants and Formation of Resistosomes

### 3.1. Plant Cell Death Signal Transmission Network

Intracellular nucleotide-binding (NB) domain and LRR-containing proteins are an essential part of plant defense and innate immune systems. NB-LRR (NLR) proteins interact with effector proteins that are synthesized by pathogens. As a result, a protective reaction occurs, often involving rapid and limited cell death at the locus of infection called the hypersensitive response site. This type of immunity is termed Effector-Triggered Immunity (ETI). Plant NLR proteins are signal transduction ATPases with numerous domains (STAND superfamily) [77,78,79,80]. NLRs can be grouped into subclasses in accordance with their N-terminal domain. CNLs contain coiled-coil domain or RPW8-type (Resistance to Powdery Mildew 8) coiled-coil (CC_R_, RNL proteins), and TNLs contain toll/interleukin-1 receptor-type domain [80,81]. RPW8 is a membrane-associated protein from *A. thaliana* [81]. Typical coiled-coil domains incorporate the motif “EDVID”, while the RPW8-like coiled-coil CC_R_ domains do not contain this motif [81]. Since in monocotyledonous plants, including cereals, NLR proteins incorporate only CC domains, one may assume that toll-like/interleukin-1 receptor-type domains were lost in the process of divergence from the common angiosperm ancestor [82]. In addition to ETI, there is another basic mechanism of plant resistance to pathogens, called pattern triggered immunity (PTI), that limits pathogen spreading but does not trigger cell death. This pathway requires participation of receptor-like kinases and/or receptor-like proteins that lack a protein kinase domain situated in the plasma membrane. There is very little evidence that PTI may be involved in viral resistance. None of the PTI receptors were shown to recognize virus particles [80].

Typical representatives of CNLs are HopZ-Activated Resistance1 (ZAR1) from *Arabidopsis* [83] and *N. benthamiana* [84] as well as Rx from potato [85]; for TNLs—*N* gene product from tobacco (*N. tabacum*) [11] and RPP1 from *Arabidopsis* [86]. Some NLRs act as singletons, combining functions of pathogen sensing and immune signaling, while others form a network of interactions involving at least two factors [87,88].

Activated singleton (for example, ZAR1) and helper NLR proteins oligomerize, forming circular “tunnel” structures termed resistosomes, consisting of several molecules of the corresponding protein or a complex of distinct proteins, and forming a Ca^2+^ channel in the plasma membrane (PM) of the cell [77,89,90,91,92]. The N-terminal helix α1 containing the MADA motif was found in about 20% of helper angiosperm CNLs capable of oligomerization [93]. In contrast, the NLR sensor proteins lack the MADA motif [93].

There are known examples in which overexpression of plant signal CC and TIR domains alone lead to the induction of HR. These include the TIR domains of factors RPS4, RPP1A, and At4g19530 in *Arabidopsis* [94], CC_R_ (RPW8-like) domains of ADR1 and NRG1 proteins in *Arabidopsis* and *N. benthamiana* [95], as well as the CC domains from wheat proteins Sr33 and Sr50 in *N. benthamiana* [96]. The N-terminal fragment of the CC domain of factor AT1G12290 (100 aa) from *Arabidopsis* was sufficient to induce HR in *N. benthamiana* but not in *Arabidopsis* [97]. The CC-domain of barley protein MLA10, capable of forming a rod-shaped homodimer, was also capable of inducing cell death in *N. benthamiana* [98]. Most likely, a homodimer is also formed via self-association of the TIR domain from flax (*Linum usitatissimum*) L6 resistance protein [99]. On the contrary, the expression of the CC domain of the NLR protein Rx from potato in *N. benthamiana* leaves did not cause HR, unlike the expression of the NB domain [100]. For ADR1 and NRG1, it was shown that HR development caused by the expression of CC_R_ domains occurred independently of SGT1, a cofactor of the chaperone HSP90 [95].

The NLR required for cell death (NRC) network in plants is comprised of at least two levels [101] and consists of sensor and helper proteins, forming pairs or genetically unlinked receptor networks [87,88,101,102]. For example, PVX CP triggers Rx activation [85] followed by signaling through any of the NRC2,3,4 CNL helper proteins and consequent cell death in *N. benthamiana* [87,103]. A similar pathway was described for the Bs2 sensor from wild pepper (*C. chacoense*) [104]. Sensor Rpi-blb2 from potato interacts with AVRblb2 effector from *Phytophtora infestans* [105] and activates NRC4, but not NRC2 [104,106]. It is important to note that the Rx sensor does not form stable complexes with either NRC2 or NRC4 and thus does not participate in their oligomerization. Activated homo-oligomers NRC2/NRC4 are localized to puncta that associate with the PM, while Rx remains in the cytosol [104]. *N. benthamiana*, in addition to genes encoding factors NRC2-NRC4, encodes atypical gene NRCX, defined as NLR modulator [107]. NRCX protein does not contain the usual N-terminal MADA motif and is not capable of oligomerization. When co-expressed, NRCX has a negative effect on NRC2 и NRC3, but not NRC4 [107]. Proteins belonging to the NRC network form an evolutionary superclade, estimated to be 100 million years old. In some cases, functionally related proteins are not phylogenetic relatives. In *Solanaceae* the NRC accounts for up to half of the NLR-ome [87]. It is assumed that, while performing the functions of pathogen perception and helper activation, in the course of evolution, sensor CNLs have lost the ability to oligomerize [104].

RNL proteins function downstream of pathogen detection proteins. For the effective manifestation of toll-like/interleukin-1 receptor-type signaling in plants lipase-like protein EDS1, other helper RNL proteins as well as Activated Disease Resistance gene 1 (ADR1) and NRG1 are needed [108,109]. Both proteins represent two subgroups of the RNL family. In *Arabidopsis*, NRG1 is required for cell death and ADR1 for transcriptional reprogramming of resistance genes [110,111,112]. In *N. benthamiana*, toll-like/interleukin-1 receptor-type (TIR) signals are transmitted primarily by NRG1 [113,114,115]. Two other lipases participating in this pathway are PAD4 and Senescence-Associated Gene 101 (SAG101). Either PAD4 or SAG101 forms functional heterodimeric subunits with EDS1 [116,117]. ADR1-EDS1-PAD4 module provides transcriptional defenses that restrict pathogen growth [118,119], whereas the NRG1-EDS1-SAG101 module is involved in both cell death and transcriptional regulation in *N. benthamiana* [112].

NLR-ID proteins contain the so-called integrated domain (ID), for example in the TNL pair RRS1/RPS4 from *Arabidopsis*, the sensor associates with a helper. The WRKY (defined by the conserved WRKYGQK motif and a zinc-finger region) transcription factor ID domain of RRS1 protein is located at the C-terminus. Interaction of the ID with the adjacent domain 4 (DOM4) retains the protein in an inactive form. Bacterial effector AvrRps4 from *Pseudomonas syringae* disrupts this interaction, leading to derepression of RRS1. NLR-IDs act in combination with the canonical NLR protein, which performs a helper function, inducing cell death [120]. Such pairs may also contain a CC domain instead of TIR, for example the pair of rice proteins RGA4/RGA5 [121].

Some NLR proteins can cause unwanted events in plants known as hybrid incompatibility, which is apparently due to the fact that incorrect interaction of sensors and helpers can cause an unintentional defense response [122]. For example, ADR1 and NRG1 proteins may recognize various “wrong” sensors, which increases the risk of autoimmune reactions [123]. The mechanisms that prevent aberrant TIR/TNL activity were reviewed recently [109]. To date, NLR genes are not classified as “lethal phenotype genes” necessary for the survival and development of plants [124].

### 3.2. Resistosomes in Plants

NLR proteins that combine the functions of sensors and helpers/executors are termed singleton NLRs. The mechanism of action of some of them is quite well studied. For example, according to cryo-electron microscopy, CNL protein ZAR1 forms a wheel-like activated pentameric membrane-associated complex termed the resistosome, which is an α-helical barrel interacting with the LRR and winged-helix domains [89,90]. In an inactive state, ZAR1 is precomplexed with the pseudokinase RKS1 (resistance-related kinase 1); during activation, the effector AvrAC from *Xanthomonas campestris* uridylates decoy kinase PBL2 (probable serine/threonine-protein kinase 2), which associates with the complex ZAR1-RKS1, causing its structural rearrangement and release of ADP from nucleotide-binding domain with subsequent oligomerization of the heterotrimer and its movement from the cytosol into the PM [90]. Structurally, the ZAR1 resistosome resembles the metazoan apoptosome APAF1 (apoptotic protease activating factor 1) [125,126,127] and inflammasome NAIP2-NLRC4 (NLR family apoptosis inhibitory protein 2—NLR family, CARD domain containing 4) [128,129,130]. Experiments involving in vitro reconstitution of ZAR1 resistosome in planar lipid bilayers showed that it is a calcium ion channel, which triggers Ca^2+^ influx in *Arabidopsis* protoplasts; it can be localized to the PM but not to the endoplasmic reticulum [131]. It was reported that activation of ZAR1 leads to perturbation of chloroplasts, abrupt disintegration of nuclei, and cell collapse [131]. CNL protein Sr35 from monocotyledonous wheat interacts with effector AvrSr35 of the wheat stem rust pathogen (fungus *Puccinia graminis* f. sp. tritici, Pgt). Activation of Sr35 occurs when its LRR domain interacts with AvrSr35. Furthermore, the Sr35-AvrSr35 complex oligomerizes, binding ATP, and forms a pentameric structure similar to “dicotyledonous” resistosome ZAR1, despite only 28.4% sequence conservation [92]. Complex Sr35-AvrSr35 was functional in *N. benthamiana*, inducing cell death. The Sr35 resistosome also exhibits Ca^2+^ channel activity in oocytes (*Xenopus laevis*) [92].

The putative pentameric resistosome NRG1 is formed from heterotrimeric subunits consisting of a helper RNL protein, interacting with cofactor lipases EDS1 и SAG101 [112]. Autoactive mutant NRG1 from *Arabidopsis* becomes localized to the PM and forms Ca^2+^-permeable cation channels [20]. Sensor TNL together with β-estradiol promotes oligomerization of NRG1-EDS1-SAG101 complex and directs it to the PM and nucleus [112].

The resistosome of another resistance factor from *Arabidopsis*—TIR-containing protein RPP1, is tetrameric. Effector ATR1 from the oomycete pathogen *Hyaloperonospora arabidopsidis* activates RPP1 through LRR domain [91]. Unlike ZAR1, the RPP1 protein in the resistosome remains bound to ADP; the effector ATR1 retains interaction with LRR. This resistosome exhibits enzymatic activity and hydrolyzes nicotinamide adenine dinucleotide (NAD+). Resistosome RPP1 is a holoenzyme formed by two asymmetric homodimers, creating active sites for NAD^+^ hydrolysis. NAD+ glycohydrolase (NADase) activity requires the proximity of two TIR domains and is enhanced in the presence of polyethyleneglycol, Mg^2+^ ions (10 mM), and, to a lesser extent, Ca^2+^ [91,132]. TNL protein ROQ1 from *N. benthamiana* also forms a tetrameric resistosome in complex with the XopQ effector (*Xanthomonas* outer protein Q) from *Xanthomonas euvesicatoria*. The LRR domain has a horseshoe shape and wraps around XopQ. The nucleotide-binding domains of the oligomers of the resistor-activated ROQ1 interact with ATP and a Mg^2+^ ion. NADase activity was suggested, but not experimentally proven. The structure of the putative active center was similar to the resistosome RPP1 [77].

### 3.3. Metazoan Apoptosomes/Inflammasomes

Central nucleotide-binding domain (NOD, nucleotide-binding oligomerization domain), phylogenetically related to plant NB-ARC, was found in such proteins as human APAF-1 and its homolog CED-4 from *Caenorhabditis elegans* [133,134]. Unlike plant NLR proteins, the C-terminal region of APAF-1 contains WD40 repeats, also known as WD (terminating in a tryptophan-aspartic acid) or beta-transducin repeats. According to phylogenetic analysis, the domain NOD of APAF-1 is more similar to plant nucleotide-binding domains than to the NOD domains of other mammals [129,135]. The N-terminal region of APAF-1, like many other metazoan NLR proteins, contains a CARD domain (caspase recruitment domain). Often, a Pyrin (PYD, also known as PAAD/DAPIN) domain is found instead of CARD. Inflammasome NAIP2-NLRC4 and apoptosome APAF-1 activate caspase-1 and caspase-9, respectively [130,136,137]. CC domains are well-defined and form an α-helical barrel but largely buried in the tunnel-shaped structure of resistosome, whereas CARD domains are completely disordered in the absence of interacting partners and remain flexible in the oligomeric structure [89,90]. The TIR domains of the RPP1 resistosome also form a stable tunnel structure [91]. Structural differences between plant resistosomes, NAIP2-NLRC4 inflammasome and APAF-1 apoptosome suggest that transmission of the cell death signal using NLR proteins in plants and animals may differ significantly, primarily due to the action of the procaspase binding domain CARD. Perhaps, the ZAR1 resistosome affects the permeability of the PM or perturbs its integrity [89,90]. The NLRC4 inflammasome activates the protein Gasdermin D (GSDMD) via caspase-1, forming membrane pores and mediating a type of programmed cell death known as pyroptosis [138]. In contrast, the RPP1 resistosome acts as a holoenzyme for NAD^+^ hydrolysis [91], resembling the mechanism of action of metazoan CARD-containing holoenzymes responsible for recruitment of procaspases. It should be noted that the TIR domains of other plant NLR proteins are also capable of NADase activity, as well as some TIR-containing animal proteins devoid of NOD and C-terminal WD40/LRR domains [132,139]. NAD cleavage by TIR domains leads to the release of adenosine diphosphate ribose from NB-ARC domains, which, in turn, activates Ca^2+^ influx to the cytosol, subsequently activating the HR [77,92]. In addition, it is assumed that NADase activity of TNL resistosomes can produce cyclic-ADP-Ribose (vcADPR) and activate the formation of “downstream” RNL resistosomes, which, in turn, cause cell death via Ca^2+^ influx [112]. Interestingly, expression of wheat CNL Sr35 together with its pathogen effector AvrSr35 in Sf21 insect cells induced cell death in the absence of other plant proteins [92].

SARM-1 (sterile alpha and TIR motif containing 1) protein induces neuronal cell death and axon degeneration related to NADase activity of the protein dimer [140]. In addition to TIR, it contains other domains, including tandem sterile alpha motifs (SAM) responsible for its self-association. Plant and animal TIR domains are phylogenetically distant but structurally similar as demonstrated by crystallization of human octameric TIR from SARM-1 [132]. In vitro experiments show that animal TIR domains hydrolyze NAD^+^ much more rapidly than those of plants [132]. Surprisingly, expression of either human TIR domain [132] or an autoactive fragment of SARM-1 [139] in *N. benthamiana* caused HR [132,139], even in EDS1 knockout plants [132], which indicates a mechanism different than that of typical plant TNL signaling [132]. A fusion protein consisting of the human SAM domain responsible for oligomerization and the plant RPS4 TIR domain was functional and induced HR in *N. tabacum* [139].

The TIR domain of RPS4 from *Arabidopsis* was successfully fused to an animal NLRC4 protein incorporating an LRR domain, which, in particular, distinguishes it from WD-containing APAF-1. Expression in *N. benthamiana* and *N. tabacum*, together with combinations of assistant genes (NAIP5/flagellin FlaA from *Legionella pneumophila* or NAIP2/rod protein PrgJ from *Salmonella typhimurium*), led to the formation of stable immunoprecipitable “plant inflammasomes” capable of inducing HR [141].

A proton pump formed by bacterio-opsin (bO) and located in the plasma membrane of transgenic *N. tabacum* cells activates HR, manifested as a “lesion mimic phenotype” [142]. Protein HrpZ1 from plant pathogen *P. syringae* generates membrane-associated pore structures in parsley (*Petroselinum crispum*) cell suspensions and activates MAPK cascades, which are involved in PCD signaling [143].

## 4. Common Features of Antiviral Resistance across the Kingdoms

Virus-induced ETI in animals is insufficiently studied. The NKG2D receptor of natural killer cells interacts with viral effectors, which leads to cell activation. One of the most striking examples is the mouse cytomegalovirus, encoding the m18 effector, which affects the deacetylation of histones in host cells. Viral infections can affect the efficiency of protein synthesis in animals, which might induce ETI as well. For example, protein factors BCL-xL and MCL-1 triggered pyroptosis associated with gasdermin E in virus-infected barrier epithelial cells [144].

Other animal pathogens may interact with NLRs either directly or indirectly. Mouse NAIP proteins can serve as examples of the first option. NAIP5 binds the bacterial protein flagellin, whereas NAIP2 binds the inner rod component from type III secretion system (T3SS). In turn, both proteins are able to activate the NLRC4 inflammasome and remain members of the heterooligomeric complex as single subunits. The HD1 and HD2 motifs of the nucleotide-binding domain NACHT (found in NAIP, CIITA, HET-E, TP1, hence the name) play a key role in such recognition, and the flanking BIR1 and LRR domains are auxiliary [145]. Mutual replacement of NACHT domains in NAIP chimeric proteins changes the specificity of recognition of their corresponding partners. Synthetic recombinant protein NAIP 2/5 provides recognition of both bacterial Avrs [146]. A direct mechanism of intracellular perception between Avrs and sensors in plants, in particular, may imply interaction of the effector with the LRR domain (ATR1 and RPP1) [147] or C-JID domain (C-terminal jelly roll/Ig-like) located downstream of LRR (XopQ and ROQ1) [148]. One can also mention recognition of PVX CP by the C-terminal LRR domain of Rx from potato.

In addition to the well-studied example of ZAR1, involved in indirect ETI recognition in plants (Section 3.2) [89,90], RIN4 (RPM1-interacting protein 4) “guardee” protein from *Arabidopsis* forms complexes with NLR guard proteins RPS2 and RPM1. If the protein recognized by the effector has no other physiological function, it is called a “decoy” [78]. Protease AvrRpt2 from *P. syringae* cleaves RIN4 and releases the guard [149]. The AvrRpm1 effector performs (ADP)-ribosylation of RIN4, which is subsequently phosphorylated [150]. Indirect recognition in animals can be represented by NLRP1b protein with noncanonical organization of domains—the C terminal CARD is separated from LRR by the so-called function-to-find domain (FIIND). Protease LF (Lethal Factor) from *Bacillus anthracis* cleaves off the C-terminus of NLRP1b capable of oligomerization and caspase recruitment [78,151]. Such a model is often defined as “recognition with integrated domains (ID)” [78]. Plant TNL guardee RRS1 interacts with the TNL helper/executor RPS4 and inhibits its autoactivity (Section 3.1, Figure 1). RRS1 interacts, in particular, with the PopP2 effector from *Ralstonia solanacearum* via C-terminal ID domain WRKY, which leads to its acetylation [78,152].

Recently, striking similarities between effector-triggered PCD pathways in plants and animals have been found. Foremostly, these concern the formation of similar membrane-associated structures, playing a key role in the induction of HR in plants and apoptosis/pyroptosis in animal cells. Such structures are called “resistosomes” [83,84,86]. Currently known plant resistosomes allow Ca^2+^ influx into the cytosol or demonstrate NADase activity [89,91].

Viral effectors can interact with NLR sensors via various mechanisms, changing their conformation and thereby activating them. Avr/NLR sensor interaction can be mediated by and involve several assistant proteins, which, as a rule, subsequently retain interaction with the NLR protein undergoing oligomerization [89,90]. The NAIP2/NLRC4 inflammasome, like many plant resistosomes, contains auxiliary proteins [128,129,130]. “Assistant” and “helper” proteins should be distinguished; the former are connected to the periphery of the resistosome, while the latter are a core part. In addition, singleton CC-containing proteins are able to form resistosomes “unaided”, whereas the sensor CNLs transmit the signal to the next level and helper CNLs form resistosomes [89,90,92,104]. Both CNLs and TNLs can form “sensor pairs”, while the subsequent steps of signal transmission remain unclear [101]. For comparison, apoptosome APAF-1 formation requires the participation of cytochrome C and dATP, and the NAIP2/NLRC4 inflammasome requires involvement of pathogen-associated molecular patterns (PAMPs) [125,126,127,128,129,130]. TNL signaling is indirect. Oligomeric TNL resistosomes display NADase activity and activate helper RNL proteins with assistant lipase-like protein EDS1 in complex with either SAG101 or PAD4 lipases. To date, it is assumed that RNL NRG1 forms resistosomes that allow Ca^2+^ influx, while RNL ADR1 acts as a transcriptional regulator [112]. Interestingly, induction of cell death in plants using animal TIR domains dispenses with EDS1 assistance [132].

At the level of the PM (Figure 1), the main similarity between plant resistosomes and apoptosomes/inflammasomes of animals is primarily structural. Nevertheless, it is necessary to note the phylogenetic relationship between the nucleotide-binding plant domain NB-ARC and the animal domain NOD of APAF-1 [135]. TIR domains of plants and animals have no significant homology at the amino acid level, but are functionally interchangeable. For example, the chimeric “plant inflammasome” that includes the plant TIR domain together with a “complete” animal NLRC4 protein demonstrated NADase activity followed by HR induction. For its formation in plant cells, as well as in animal cells, auxiliary NAIP and PAMP proteins were needed [141]. From the point of view of modular organization of cell death pathways, inflammasome-inducing pathogen signal PAMP resembles the Sr35 singleton resistosome [92]. A fusion protein consisting of human oligomerization domain SAM and plant TIR was functional and provided activation of the hypersensitive response in plants [139]. CC and CC_R_ domains are present only in plants. On the contrary, the CARD domain, which provides activation of procaspases, is absent in plant NLR proteins. The mechanism of activation of plant metacaspases and caspase-like proteins involving membrane Ca^2+^ channels need to be studied in more detail (Figure 1).

Plants and animals share proteins containing NBD and LRR. It is believed that these proteins evolved from a common prokaryotic ancestral ATPase. NBDs are highly conserved but, based on phylogenetic analysis, it is generally accepted that they diverged early from the NB-ARC (plants, partly animals) and NACHT (animals) lineages [78,135]. Fungi also possess both types of NBD but lack LRRs [153]. Either NB-ARC or NACHT domains probably served as “assembly centers”, which developed independently from each other, collecting sets of N- and C-terminal functional domains that differ significantly in plants and animals [78,154]. In both kingdoms, the system of triggering and NLR signaling for PCD pathways is very complex and flexible. Different types of Avr recognition, various guardee, decoy, helper, and assistant proteins together with homo- and heterooligomeric membrane protein complexes present a wide field for research. The structural features of NLRs allow obtaining functional fluorescent fusion proteins or separate domains. Additional domains such as animal BIR or plant integrated domains involved in Avr recognition as well as chimeric proteins including TIR [139,141] and NACHT [146] domains look like promising targets for NLR engineering. In our view, the number of similar or interchangeable elements in the organization of plant and animal innate immune systems seems sufficient to raise the question posed in the title of this review—is there a common mechanism of antiviral resistance conserved across kingdoms? The answer can only be obtained experimentally.

## 5. Conclusions

The development of PCD in plants and *Metazoa* differs significantly. However, recent structural studies of plant resistosomes have revealed many similarities with animal apoptosomes and inflammasomes. It should be recognized that there are obvious blind spots in our understanding of the relationship between the formation and functioning of resistosomes, the presence of caspase-like activity in plants, and the development of the HR. Nevertheless, it is necessary to mention that the progress made to date is considerable. The relative simplicity and diversity of virus-specific effectors of the hypersensitive response makes them easier to study in comparison with effectors of fungal and bacterial pathogens. In addition to the transient expression of effector genes using agroinfiltration, it is possible to quickly obtain stable transgenes (*Arabidopsis*) and infect plants with natural viruses or viral vectors. Special attention should be paid to the data on the induction of HR during the expression of individual domains of plant NLR proteins. Further study of such effects may have practical significance for agriculture and medicine.

## 6. Future Directions

The multi-domain organization of NLR signaling proteins in plants and their functional homologues in *Metazoa*, as well as the variety of corresponding pathways and a “modular” structure of interaction between effector, sensor, and helper proteins allows for the envisioning of various “cross-kingdom” studies. In this regard, we can mention modeling of cell death pathways in plants and animals, creation of chimeric plant/animal NLR proteins, and assembly of resistosome-like structures in vitro (in planar lipid bilayers) or in vivo in plant and animal cells, including heterogeneous plant/animal subunits.

## Figures and Tables

**Figure 1 ijms-24-13625-f001:**
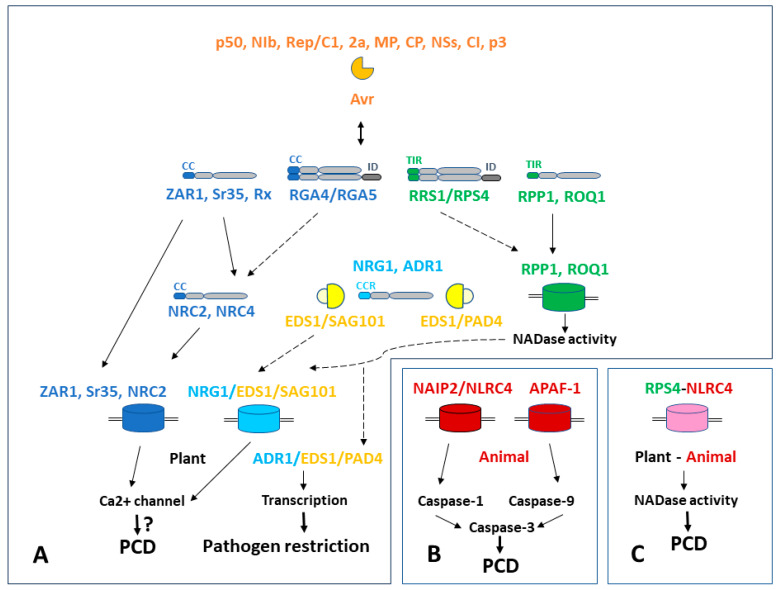
Schematic representation of plant NLR signaling and resistosome formation alongside the animal apoptosome and inflammasome, and the main programmed cell death (PCD) pathways. (**A**) Plant signaling network. CNL, RNL, TNL pathways are shown in blue, pale blue, and green, respectively. Avr effectors are displayed in orange and helper lipases are displayed in yellow. (**B**) Animal apoptosome (APAF-1) and inflammasome (NAIP2/NLRC4) are shown in red. (**C**) Chimeric “plant inflammasome” generated with fusion protein consisting of plant TIR domain and animal NLRC4 subunit is shown in purple. Dotted lines demonstrate the intended pathways.

**Table 1 ijms-24-13625-t001:** Viral effectors inducing hypersensitive response in plants **^a^**.

Type of Virus Specific Proteins Dealing with Programmed Cell Death	Virus Protein	Virus Protein/Characteristics	Type NLR (Nucleotide-Binding Leucine-Rich Repeats Proteins)	Ref.
Replication-related	P50 Nib RdRp Rep/C1 2a	TMV PepMoV PlAMV TYLCV CMV	TIR-NBS-LRR CC-NBS-LRR CC-NBS-LRR Not reported type NLR TIR-NBS-LRR	[11,12,13,14,15,16,17,18,19,20,21] [22,23,24,25] [26,27] [28,29,30,31] [32]
CP	CP P38 CP CP	PVX TCV PVY CMV	CC-NBS-LRR CC-NBS-LRR TIR-NBS-LRR CC-NBS-LRR	[33,34,35,36,37] [38,39,40,41,42,43] [44,45,46,47] [48,49]
MP	MP MP (29K) NSM P6	ToMV TRV TSWV CaMV	CC-NBS-LRR Not reported type NLR CC-NBS-LRR Not reported type NLR	[50,51,52,53] [54] [55,56,57,58] [59,60,61,62,63]
Other proteins	NSs (silencing suppressor) AC4 (silencing suppressor) C1 (cylindrical protein) P3 (multifunctional protein) HC-Pro (multifunctional protein)	TSWV ToLCNDV SMV	CC-NBS-LRR CC-NBS-LRR CC-NBS-LRR	[64,65,66,67,68,69] [70,71] [72,73,74,75,76]

**^a^** Abbreviations: CP, coat protein; MP, movement protein; TIR, toll-like/interleukin-1 receptor-type; CC, coiled coil.

## Data Availability

Not applicable.

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
