# Peer review of "Signaling and Resistosome Formation in Plant Innate Immunity to Viruses: Is There a Common Mechanism of Antiviral Resistance Conserved across Kingdoms?"

_ijms, 2023, doi:10.3390/ijms241713625_

Round 1

Reviewer 1 Report (New Reviewer)

The paper draws interesting parallels between the mechanisms of innate immunity in plants and animals. As it is based on a large amount of experimental data and modern concepts, this paper would be useful for anyone interested in virus-host interactions. Therefore, in my opinion, the paper is worth publishing. However, I have some concerns that should be addressed before the paper can be accepted for publication.

The text has many problems. Some text fragments are not directly related to the main topic of the article or are overloaded with unnecessary information. In many cases, the internal logic of the text is distorted. Some sentences are difficult to understand due to internal contradictions or problems with the English language.

I have tried to correct some sentences (see comments below), but the work that authors should do on the text should not be limited to these corrections. The entire text should be critically revised so that (1) the meaning of each sentence becomes clear and (2) the logic is maintained from beginning to end.

Major concerns

The title - I suggest rephrasing:

Signaling and resistosome formation in plant innate immunity to viruses: is there a common mechanism of antiviral resistance conserved across kingdoms?

The authors seem to misunderstand the term "effector". They think that any protein that induces HR can be called an effector. This is incorrect because effectors are proteins that are delivered by pathogens into host cells to alter host metabolic processes in favor of the pathogen. Therefore, not all effectors cause HR, and not all HR-inducing proteins are effectors. The use of the word "effector" should be corrected throughout the text.

Section 2 should be shortened to a few sentences since all of this information is presented in Table 1. These phrases can start Section 3, which should be renamed accordingly.

Sections 3.1, 3.2, and 3.3 should be combined and greatly condensed. The information contained in this section can be found in numerous reviews, so only the facts necessary for further discussion should be included here.

Section 4, the most important part of the paper, should be reorganized to make the point clearer - see my specific comments below.

Other points

Line 17. protein effectors - should be ‘proteins’

Line 18. hypersensitive response - It should be said that hypersensitive response is a type of cell death, otherwise there is no link to the next sentence. The abbreviation HR should be introduced here. 

Line 19. the effector - should be ‘HR-inducing proteins’

Line 25. What is NADase?

Line 35. a number of fundamental basic similarities

The sentence is unclear because it does not say which objects have these similarities.

Line 44. Why alternatively? To what is it an alternative?

Line 49 - Typo: termed_the

Line 51 has remained - should be ‘remains’

Line 58 - allow for the localization - enable local mounting of HR? Besides, how cell degradation can upregulate expression of defense genes? The phrase is unclear and should be re-written.

Lines 61-70. A circular paragraph. The first and last sentences are almost identical.

Line 241. Should be ‘domains’

Line 241 and hereafter: nucleotide-binding LRR protein - usually nucleotide-binding is abbreviated to NB, and the term NB-LRR is used, which is more common 

Line 246. NLR-dependent immunity is termed Effector-Triggered Immunity (ETI).

Incorrect statement. 

Line 355. The term ‘resistosome’ should be defined.

Lines 362-263. Structurally, the ZAR1 resistosome resembles the metazoan apoptosome APAF1 [125-127] and inflammasome NAIP2-NLRC4 [128-130].

Should be moved to the Section 3.5 or Section 4. Here the comparison to APAF1 and inflammasome looks inappropriate.

Section 3.5 should be renamed to ‘Metazoan apoptosomes/inflammasomes’, as no comparison to plants is presented here.

Lines 461-463. The phrase should be re-written.

Section 4:

Lines 460-481. These general considerations should be moved to the Introduction or to the sections above.

Lines 504-510 (RIN4 story). Inappropriate here. Adds nothing, so can be easily deleted.

Line 520. The first phrase should be deleted.

Line 533. What is the link between the sentence ending with [82] and the previous and next sentences? It should be deleted. After that, this paragraph should be moved to the section describing animal systems, as no comparison to plants is given here. 

Line 577. induction of - can be omitted

Lines 576-580. This paragraph is unrelated to the text fragments below and above. May be moved to Conclusions.

Lines 593-595. The results of ETI induced in plants by viral effectors and capable of functioning as a powerful yes/no “switch” are relatively easy to observe and analyze. The structural features of NLRs allow obtaining functional fluorescent fusion proteins or separate domains. 

Unclear why this is mentioned here. Should be deleted.

Fig. 1 - I suggest to frame the panel A as it is done for B and C.

Moderate corrections of the English language are required.

Author Response

The paper draws interesting parallels between the mechanisms of innate immunity in plants and animals. As it is based on a large amount of experimental data and modern concepts, this paper would be useful for anyone interested in virus-host interactions. Therefore, in my opinion, the paper is worth publishing. However, I have some concerns that should be addressed before the paper can be accepted for publication.

The text has many problems. Some text fragments are not directly related to the main topic of the article or are overloaded with unnecessary information. In many cases, the internal logic of the text is distorted. Some sentences are difficult to understand due to internal contradictions or problems with the English language.

I have tried to correct some sentences (see comments below), but the work that authors should do on the text should not be limited to these corrections. The entire text should be critically revised so that (1) the meaning of each sentence becomes clear and (2) the logic is maintained from beginning to end.

Re: Thank you for your thorough review of our manuscript. We appreciate your valuable feedback and concerns. Please find the point by point responses to your concerns below (appropriate changes within the manuscript were performed in ‘track changes’ mode). An extra English editing by the native speaker was performed.

Major concerns

The title - I suggest rephrasing:

Signaling and resistosome formation in plant innate immunity to viruses: is there a common mechanism of antiviral resistance conserved across kingdoms?

RE: Thank you. The title was changed accordingly.

The authors seem to misunderstand the term "effector". They think that any protein that induces HR can be called an effector. This is incorrect because effectors are proteins that are delivered by pathogens into host cells to alter host metabolic processes in favor of the pathogen. Therefore, not all effectors cause HR, and not all HR-inducing proteins are effectors. The use of the word "effector" should be corrected throughout the text.

RE: The text was carefully checked and modified as suggested.

Section 2 should be shortened to a few sentences since all of this information is presented in Table 1. These phrases can start Section 3, which should be renamed accordingly.

RE: Section 2 was significantly shortened.

Sections 3.1, 3.2, and 3.3 should be combined and greatly condensed. The information contained in this section can be found in numerous reviews, so only the facts necessary for further discussion should be included here.

RE: Sections 3.1, 3.2, and 3.3 were combined, renamed as 3.1 “Plant cell death signal transmission network” reorganized and shortened. 

Section 4, the most important part of the paper, should be reorganized to make the point clearer – see my specific comments below.

RE: Please see below.

Other points

Line 17. Protein effectors – should be ‘proteins’

RE: Corrected.

Line 18. Hypersensitive response – It should be said that hypersensitive response is a type of cell death, otherwise there is no link to the next sentence. The abbreviation HR should be introduced here.

RE: The first sentence in the Abstract was changed: “Virus-specific proteins, including coat proteins, movement proteins, replication proteins and suppressors of RNA interference are capable of causing the hypersensitive response (HR) that is a type of cell death in plants”.  

Line 19. The effector – should be ‘HR-inducing proteins’

RE: Corrected.

Line 25. What is NADase?

RE: Systematic name NAD+ glycohydrolase was added to the text.

Line 35. A number of fundamental basic similarities

The sentence is unclear because it does not say which objects have these similarities.

RE: The sentence was rephrased: “At the morphological and biochemical levels, there are a number of basic  similarities, even among representatives of different kingdoms of living organisms (plants and animals), although the molecular mechanisms of programmed cell death vary”.

Line 44. Why alternatively? To what is it an alternative?

RE: “Alternatively” was deleted.

Line 49 – Typo: termed_the

RE: Corrected.

Line 51 has remained – should be ‘remains’

RE: Corrected.

Line 58 – allow for the localization – enable local mounting of HR? Besides, how cell degradation can upregulate expression of defense genes? The phrase is unclear and should be re-written.

RE: The phrase was deleted from the text.

Lines 61-70. A circular paragraph. The first and last sentences are almost identical.

RE: The last sentence was rewritten: “Information concerning numerous viral proteins involved in triggering HR is summarized in Table 1”.

Line 241. Should be ‘domains’

RE: The term is used in the plural.

Line 241 and hereafter: nucleotide-binding LRR protein – usually nucleotide-binding is abbreviated to NB, and the term NB-LRR is used, which is more common

RE: “Nucleotide-binding LRR” was changed to NB-LRR throughout the text.

Line 246. NLR-dependent immunity is termed Effector-Triggered Immunity (ETI).

Incorrect statement.

RE: The phrase was modified: “This type of immunity is termed Effector-Triggered Immunity (ETI)”.

Line 355. The term ‘resistosome’ should be defined.

RE: The sentence was changed: “according to cryo-electron microscopy, CNL protein ZAR1 forms a wheel-like activated pentameric membrane-associated complex called resistosome, which is an α-helical barrel interacting with the LRR and winged-helix domains”.

Lines 362-263. Structurally, the ZAR1 resistosome resembles the metazoan apoptosome APAF1 [125-127] and inflammasome NAIP2-NLRC4 [128-130].

Should be moved to the Section 3.5 or Section 4. Here the comparison to APAF1 and inflammasome looks inappropriate.

RE: This resemblance was specifically emphasized in the initial “plant” publications. Therefore, from our point of view, it makes sense to keep this mention here as well. In Section 3.2, just one sentence is devoted to this similarity.

Section 3.5 should be renamed to ‘Metazoan apoptosomes/inflammasomes’, as no comparison to plants is presented here.

RE: Section was renamed according to Reviewer’s comment.

Lines 461-463. The phrase should be re-written.

RE: Now this phrase looks like this: “Viral infections can affect the efficiency of protein synthesis in animals, which might induce ETI as well”.

Section 4:

Lines 460-481. These general considerations should be moved to the Introduction or to the sections above.

RE: This fragment was partially deleted and partially moved as suggested.

Lines 504-510 (RIN4 story). Inappropriate here. Adds nothing, so can be easily deleted.

RE: RIN4 story illustrates important “guardee” and “decoy” models of indirect ETI recognition in plants. Both terms are mentioned in the text below. In our opinion, it makes sense to keep this text fragment unchanged.  

Line 520. The first phrase should be deleted.

RE: The sentence was deleted

Line 533. What is the link between the sentence ending with [82] and the previous and next sentences? It should be deleted. After that, this paragraph should be moved to the section describing animal systems, as no comparison to plants is given here.

RE: The phrase was deleted. In our opinion, it makes sense to keep this paragraph in its place since comparisons with plants are present in the text. Besides animal proteins, apoptosomes and inflammasomes, TNL, CNL, RNL, resistosomes are mentioned. 

Line 577. Induction of – can be omitted

RE: Corrected.

Lines 576-580. This paragraph is unrelated to the text fragments below and above. May be moved to Conclusions.

RE: The paragraph was moved to Conclusions.

Lines 593-595. The results of ETI induced in plants by viral effectors and capable of functioning as a powerful yes/no “switch” are relatively easy to observe and analyze. The structural features of NLRs allow obtaining functional fluorescent fusion proteins or separate domains.

Unclear why this is mentioned here. Should be deleted.

RE: The sentence was deleted.

Fig. 1 – I suggest to frame the panel A as it is done for B and C.

RE: Panel A was framed as suggested by the Reviewer.

Reviewer 2 Report (New Reviewer)

I think that this manuscript is written too poorly for an International audience.  Certainly, in the first half of the manuscript, sentences are organised in a sequence that is not coherent.  In its current state, this manuscript is not ready for submission and seems to be still in a draft form.  

It needs extensive editing for English language use. In some sections, it is not clear what point the authors are making, but rather a list of statements in an order that leaves the reader trying to work out what the take-home message is.  As such, it is very difficult to review.

  The authors describe known plant viral proteins that elicit the hypersensitive response in resistant hosts. They describe the different classes of R-gene involved and, briefly, what is known about the signal transduction pathways leading to cell death. Finally, the authors describe parallels with animal cell death.   I think the content and length of the review would be of interest.  Programmed cell death similarities/differences between animal and plant cells is often a topic of reviews, and justifiably as it is field of active research.  These authors have focussed explicitly on plant viruses as elicitors of cell death, and I think this would be of use, particularly to plant virologists, but also plant pathologists in general and any scientist interested in programmed cell death.   However, this manuscript is not well written (as the English is hard to follow), and as it stands, would not be useful for anyone looking to understand this topic quickly).

Extensive editing for English and readability is required.

Author Response

I think that this manuscript is written too poorly for an International audience.  Certainly, in the first half of the manuscript, sentences are ocusedd in a sequence that is not coherent.  In its current state, this manuscript is not ready for submission and seems to be still in a draft form.  

It needs extensive editing for English language use. In some sections, it is not clear what point the authors are making, but rather a list of statements in an order that leaves the reader trying to work out what the take-home message is.  As such, it is very difficult to review.

  The authors describe known plant viral proteins that elicit the hypersensitive response in resistant hosts. They describe the different classes of R-gene involved and, briefly, what is known about the signal transduction pathways leading to cell death. Finally, the authors describe parallels with animal cell death.   I think the content and length of the review would be of interest.  Programmed cell death similarities/differences between animal and plant cells is often a topic of reviews, and justifiably as it is field of active research.  These authors have ocused explicitly on plant viruses as elicitors of cell death, and I think this would be of use, particularly to plant virologists, but also plant pathologists in general and any scientist interested in programmed cell death.   However, this manuscript is not well written (as the English is hard to follow), and as it stands, would not be useful for anyone looking to understand this topic quickly).

RE: Thank you for your thorough review of our manuscript. We appreciate your valuable feedback and concerns. The manuscript was renamed, reorganized, intensely revised and significantly shortened (please see the Responses to Reviewer 1 as well). An extra English editing by the native speaker was performed. 

Round 2

Reviewer 1 Report (New Reviewer)

After all corrections and improvements have been made, the manuscript can be accepted for publication in its current form.

Minor editing of English language required

This manuscript is a resubmission of an earlier submission. The following is a list of the peer review reports and author responses from that submission.

Round 1

Reviewer 1 Report

This review about viral triggering innate immunity and plant cell death poses a question in the title, for which in the end, the authors did not really drawn an answer to. In addition, there is much mentioned that also comes from resistance to bacteria, fungi and oomycetes, and not just plant viruses, as stated in the title, although the set out of the subsections is largely based on what R genes in plants respond, in what ways, involving what elicitors, after infection with viruses expressing specific Avr factors. There is much redundancy between sections and it is clear that the authors either have not properly edited the manuscript, or do not know the rules of orthography of writing in English. The work needs major editing before it can be accepted and should be written more concisely such that it does not repeat the same concepts or specific details more than one.

This reviewer is not going to go over every explicit writing problem, but will point out the types of issues occurring multiple times that need fixing. First, however, it is necessary to state the rules that need to be followed:

1. When a term is abbreviated in a body of work (the Abstract is separate, but the same rules apply), it is abbreviated at the first use of the term. Here, abbreviations sometimes are not defined until long into the work, such as “NLR”. This is used a number of times, both in the text and Table 1, before it is defined in the text. [It was also defined in the Abstract, but that is considered separate from the body of the text.] The same with HSP90, MAPK, WIPK, SIPK, WRKY and EDS1, among many others.

2. After the first use of the abbreviated term, only the abbreviation is used; the expanded term is no longer used. Here, there are multiple examples of switching back and forth. The abbreviation should not be redefined in separate sentences, paragraphs or sections. Probably the most egregious example of this problem is seen with the defining of the “hypersensitive response” as “(HR)”. This occurs 10 times! There are many other terms abbreviated over and over again, especially Avr, which sometimes stands for avirulence protein and other times stands for avirulence factor, as well as “CC-NBS-LRR (CNL)”, although one time “(CNL)” is given as the abbreviation for “CC-NLR-LRR” (ln 219).

3. A term has to be used at least three times in the appropriate body (i.e., Abstract separate from text body, separate from Tables), to be abbreviated at first use. This is often not the case. For example, in the Abstract, PCD, CC, and TIR are defined, but used only once, once, and twice, respectively. There are other examples in the body of the text.

4. Plant virus names are only written in italics with a capital first letter, if used to describe virus taxonomy; i.e., an abstract entity. The names of real viruses, molecular parasites with real properties (e.g., encapsidation, replication, movement, transmission) and roles/functions, are written in Roman type without a first capital initial, unless a proper noun (e.g. Murray River encephalitis virus). Here, there is a mixture of both.

5. The names of plant species is written in italics. Here, there again is a mixture of both italics and Roman type for species names.

6. The authors state that PCD in plants is not the same as apoptosis in animals, which has specific involvement of caspases, not found outside the animal kingdom (ln 34-36). [This is a common view.] However, when discussing the HR, when they refer to the role of ROS, they refer to this process as “HR-induced apoptosis” (ln 64).

7. The Rsv1 gene is not a TNL, but an CNL (ln 56-57 vs. ln 276-281).

8. The interactions of R gene products with effectors are not just direct or via other “intermediate host factor facilitates the interactions” (ln 58), but as mentioned elsewhere, sometimes the host factor that interacts with effectors is a guard that releases the R gene product and allows it to signal a response.

9. The second “2” in “Tm22” (ln 219) should be a superscript, as given elsewhere.

10. On ln 240, a “for” should be inserted between “responsible” and “non-necrotic”.

11. The tomato Sw-5a gene (ln 262) is a paralog of the S. peruvianum Sw-5b gene (ln 228-230).

12. On ln 331, rephrase to “The NLR required for cell death (NRC) network…”.

13. On ln 332, delete the abbreviations “NLR-S” and “NLR-H”, since they were never used again, but rather either “sensor-NLR” and “helper-NLR” were used, or just “sensor” and “helper”.

14. On ln 403, change “28,4%” to”28.4%”.

15. On ln 407 and 430, a Cyrillic letter similar to the Greek capital letter Eta is given, presumably in place of “and”.

16. The sentence on ln 429-432 is not a complete sentence. It begins with “One should mention…” but does not say with regard to what!

17. The information in ln 529-534, restates what was given on ln 80-88 and elsewhere. It is not necessary to summarize this again here.

18. On ln 556, change :C-nerminal” to “C-terminal”.

19. ln 571-574 again recapitulates information stated previously.

20. “NTD” (ln 521) is just one of numerous terms not defined.

21. In some paragraphs, all the references are given at the top and then a series of facts is described with no references given. The references should appear with the facts they cite.

22. In some other paragraphs, information is given with no citation. This also needs to be corrected.

Shown in the comments above.

Author Response

please see the file attached

Reviewer 2 Report

The manuscript titled tried to summarize the current understanding of plant antiviral effector-triggered immunity (ETI) and then compare it in parallel with the apoptosis of animal. There are lots excellent reviewer papers about plant immunity, e.g., 10.1093/plcell/koac041, 10.3389/fmicb.2022.1018504, 10.1016/j.tplants.2019.09.008, 10.1146/annurev-arplant-050213-040012. However, a reviewer only focus on antiviral ETI is still lacking. However, I am totally not satisfied by the current form of the paper:

1, there are lots of avr and antiviral NLR are not cited. For instance, soybean mosaic virus and bean common mosaic virus resistance genes in soybean and common bean, resistance genes of cucumber mosaic virus, etc. Therefore, I will suggest the author just focus on those NLRs that have been cloned.

2, the general introduction of plant innate immunity is lacking. In general, people believe that the plant innate immunity is divided into two layers, the PAMP-triggered immunity (PTI) and ETI. Although there is almost no clue that PTI also contributes viral resistance, it is still need to be mentioned briefly.

3, The discussion of the common mechanism of ETI across kingdoms lacking depth.

Other minor comments:

The second paragraph of section 2 is more suitable for the section 1 introduction since it is the overall summary of HR not just about viruses.

Line 94, please consider these two references: Li et al. 2018 (10.1016/j.plantsci.2018.02.010) and Zhou et al. 2018 (10.1016/j.plantsci.2017.11.002).

Section 2.3, please consider this reference Yan et al. 2021 (10.1111/mpp.13115), which described the movement protein of tomato brown rugose fruit virus can overcome Tm-22-mediated resistance.

Please move section 3.4 before section 3.2 for the clear logic and hierarchy.

There are lots of grammatic or other language issues need to be checked, for example:

Line 23, “Ca2+ ion channel” and “NADase enzymatic activity” were suggested to be changed to “Ca2+ channel” and “NADase activity”.

Please check the usage of abbreviation, e.g., the “hypersensitive response” in lines 60, 62, 68, 101, etc should be replaced by “HR”.

Line 126, please replace “Nia” with “NIa”.

The title of section 2.1.2 may be changed to “Potyviral RdRp (NIb)” due to the description of the relationship between several potyviruses NIb and HR in the section.

Author Response

please see the file attached

Round 2

Reviewer 2 Report

Although marks in the manuscript affected my reading significantly, it is glad to see some improvement of the manuscript. However, in my opinion, I still get a clear answer to the question asked by the authors in the title. The manuscript spent lots of space talk about the combinations of Avr-NLR but not on the commons mechanism of antiviral ETI cross kingdoms. Thus, I would like to give one more chance for the authors to review this topic in a more detailed context.

Moderate editing of English language still required.

Author Response

Although marks in the manuscript affected my reading significantly, it is glad to see some improvement of the manuscript. However, in my opinion, I still get a clear answer to the question asked by the authors in the title. The manuscript spent lots of space talk about the combinations of Avr-NLR but not on the commons mechanism of antiviral ETI cross kingdoms. Thus, I would like to give one more chance for the authors to review this topic in a more detailed context.

We greatly appreciate the reviewer for the comments and one more opportunity to make corrections. To address the concern, we added more detailed discussion of virus-induced ETI (please see lines 457-491), and, in attempt to answer the question from the title, expanded the last part of chapter 4 (please see the lines 538-562). Moreover, extensive English editing was performed by native the speaker throughout the hole text as requested.